# The Microtubule Destabilizer Eribulin Synergizes with STING Agonists to Promote Antitumor Efficacy in Triple-Negative Breast Cancer Models

**DOI:** 10.3390/cancers14235962

**Published:** 2022-12-02

**Authors:** Leila Takahashi-Ruiz, Charles S. Fermaintt, Nancy J. Wilkinson, Peter Y. W. Chan, Susan L. Mooberry, April L. Risinger

**Affiliations:** 1Department of Pharmacology, University of Texas Health Science Center San Antonio, San Antonio, TX 78229, USA; 2Mays Cancer Center, University of Texas Health Science Center San Antonio, San Antonio, TX 78229, USA

**Keywords:** triple-negative breast cancer, eribulin, cGAS-STING, microtubule-targeting agents, microtubule destabilizers, type I interferons, IFNβ, cancer immunotherapy

## Abstract

**Simple Summary:**

Microtubule-targeting agents are a class of chemotherapeutics used to treat triple-negative breast cancer (TNBC). These include drugs such as eribulin that depolymerize microtubules, as well as microtubule stabilizing taxanes such as paclitaxel. TNBC is a relatively immunogenic form of breast cancer and one strategy to further increase immunogenicity is through activation of the cGAS-STING innate immune sensing pathway. We demonstrate that eribulin enhances interferon production in combination with STING agonists undergoing clinical development in immune and TNBC cells. This phenotype is shared with other microtubule destabilizers, but not with stabilizers, suggesting a mechanism downstream of microtubule depolymerization. We determined that eribulin also enhanced immunogenic responses in vivo and improved antitumor efficacy in a spontaneous murine mammary tumor model when combined with a STING agonist. These findings highlight important mechanistic distinctions between microtubule targeted chemotherapeutic drugs and provide an opportunity to improve the efficacy of immunotherapies using a compound currently used in TNBC.

**Abstract:**

Eribulin is a microtubule destabilizer used in the treatment of triple-negative breast cancer (TNBC). Eribulin and other microtubule targeted drugs, such as the taxanes, have shared antimitotic effects, but differ in their mechanism of microtubule disruption, leading to diverse effects on cellular signaling and trafficking. Herein, we demonstrate that eribulin is unique from paclitaxel in its ability to enhance expression of the immunogenic cytokine interferon beta (IFNβ) in combination with STING agonists in both immune cells and TNBC models, including profound synergism with ADU-S100 and E7766, which are currently undergoing clinical trials. The mechanism by which eribulin enhances STING signaling is downstream of microtubule disruption and independent of the eribulin-dependent release of mitochondrial DNA. Eribulin did not override the requirement of ER exit for STING activation and did not inhibit subsequent STING degradation; however, eribulin significantly enhanced IRF3 phosphorylation and IFNβ production downstream of the RNA sensing pathway that converges on this transcription factor. Additionally, we found that eribulin enhanced the population of activated CD4^+^ T-cells in vivo when combined with either a STING agonist or tumor, demonstrating the ability to function as an immune adjuvant. We further interrogated the combination of eribulin with ADU-S100 in the MMTV-PyVT spontaneous murine mammary tumor model where we observed significant antitumor efficacy with combination treatment. Together, our findings demonstrate that microtubule targeted chemotherapeutics have distinct immunological effects and that eribulin’s ability to enhance innate immune sensing pathways supports its use in combination with immunotherapies, such as STING agonists, for the more effective treatment of TNBC and other malignancies.

## 1. Introduction

Microtubule-targeting agents (MTAs) are some of the most widely used and effective chemotherapeutic agents used for the treatment of both solid tumors and hematological malignancies [1]. In particular, these drugs are a mainstay in the treatment of triple-negative breast cancer (TNBC), partly due to the lack of targeted treatment options for this aggressive and heterogenous disease. Although the clinical efficacy of MTAs was originally proposed to be due to the shared antimitotic effects of these drugs, there is increasing evidence that their diverse effects on the non-mitotic roles of microtubules in cellular trafficking and signaling contributes to their anticancer efficacy, particularly in solid tumors [1]. There are two main classes of MTAs: microtubule stabilizing drugs, including the taxanes that promote the bundling of interphase microtubules and destabilizers, such as the vinca alkaloids and eribulin, which initiate microtubule loss [1]. Drugs from each of these mechanistically distinct classes are used for the treatment of metastatic TNBC; however, there is currently no molecular rationale for the choice among these drugs either as single agents or in combination with other therapeutics. The identification of clinically relevant differences in the effects of MTAs on non-mitotic signaling and trafficking pathways that are known to be important for antitumor efficacy could have an immediate impact on the more rational use of these mechanistically underappreciated drugs in the clinic.

Some of the most important and timely non-mitotic effects ascribed to MTAs include their ability to regulate innate immune sensing pathways. Paclitaxel is unique from destabilizers, and even the structurally similar stabilizer docetaxel, in its ability to activate NF-kB-dependent production of proinflammatory cytokines downstream of TLR4 activation [2]. The ability of paclitaxel to activate TLR4-dependent immune signaling has been shown to contribute to its antitumor efficacy [3], but has also been associated with acquired drug resistance [4] and side effects such as peripheral neuropathy [5]. We recently demonstrated that the microtubule destabilizer eribulin is distinct from paclitaxel in its ability to induce a small but significant activation of the cGAS-STING innate DNA-sensing pathway through the release of mitochondrial DNA, independent of mitotic accumulation [6]. This finding is significant given the role of the STING pathway in promoting antitumor efficacy and could contribute to the increased immune infiltration seen in eribulin-treated tumors [7], as well as the favorable immunological changes observed in metastatic TNBC patients treated with eribulin [8].

The ability of eribulin to promote an immunogenic tumor environment is particularly significant in TNBC, which is the subtype of breast cancer that is most likely to respond to immunotherapy [9]. One strategy to improve the immunogenicity of tumors is through pharmacological activation of the STING pathway [10,11], which promotes the expression of type I interferons and ultimately leads to T-cell mediated tumor regression. The endogenous STING ligand, 2′,3′-cGAMP, is produced from cytosolic double-stranded DNA by cyclic GMP-AMP synthase (cGAS) [12]. When ER-resident STING binds cGAMP, it undergoes a conformational change that allows trafficking through the ERGIC/Golgi where it recruits TANK-binding kinase 1 (TBK1), which itself promotes the phosphorylation and nuclear translocation of the IRF3 transcription factor to mediate the expression of type I interferons [13,14,15]. Targeting the interferon beta (IFNβ) pathway promotes a proinflammatory microenvironment [16,17,18], improved adaptive antitumor immune response [19], suppression of cancer stem cells [20,21,22], and is correlated with improved prognosis in TNBC patients [20]. Although the endogenous cGAMP ligand has demonstrated antitumor efficacy, its relative instability and low cell permeability necessitated intratumoral administration, which has led to the development of STING agonists with more favorable pharmacological properties. These include more stable and lipophilic cyclic-dinucleotide compounds, such as ADU-S100 and E7766, as well as non-nucleotide compounds such as the orally bioavailable small molecule MSA-2, which are in active clinical development [23,24,25,26,27].

Herein we show that eribulin, in combination with diverse STING agonists, synergistically enhances the secretion of IFNβ downstream of microtubule depolymerization. This phenomenon occurs in both immune and TNBC cells and does not override normal STING trafficking or depend on the ability of eribulin to promote mitochondrial DNA release. In contrast, the ability of eribulin to enhance STING signaling appears to be through the regulation of TBK1-IRF3 activation, which allows eribulin to also enhance RNA-sensing pathways that converge on this activation. The ability of eribulin to function as a general immune adjuvant at the intersection of multiple nucleic acid immune sensing pathways in a manner distinct from the taxanes provides a mechanistic rationale for the use of eribulin and potentially other microtubule destabilizing chemotherapeutics, in combination with STING agonists and other immunotherapies in the treatment of TNBC.

## 2. Materials and Methods

### 2.1. Cells and Reagents

THP-1, HCC1937, HCC1806, MDA-MB-453, 4T1, and RAW 264.7 cells were obtained from American Type Culture Collection (ATCC, Manassas, VA, USA) and have been previously described [6,28,29]. THP-1, HCC1937 and HCC1806, and 4T1 cells were maintained in RPMI 1640 medium (Corning, Corning, NY, USA) supplemented with 10% FBS and 50 μg/mL gentamycin. MDA-MB-453 were maintained in IMEM medium (Gibco, Grand Island, NY, USA) supplemented with 10% FBS and 50 μg/mL gentamycin. RAW 264.7 cells were maintained in DMEM medium (Gibco, Grand Island, NY, USA) supplemented with 10% FBS and 50 µg/mL gentamycin. All cells were grown inside an incubator kept at 37 °C and 5% CO_2_ and routinely tested negative for mycoplasma contamination using the MycoAlert PLUS detection kit (Lonza, Basel, Switzerland), as well as regular DAPI imaging. Cell lines were validated by STR profiling (Genetica/Labcorp, Burlington, NC, USA). THP-1 Rho^0^ cells were generated as described previously [6]. Pharmacological reagents used in this study include eribulin (Eisai Inc., Woodcliff Lake, NJ, USA), vinorelbine (AdooQ Biosciences, Irvine, CA, USA), combretastatin A-4 (Sigma Aldrich, St. Louis, MO, USA), paclitaxel (Sigma Aldrich), HT-DNA (Sigma Aldrich), cGAMP (InvivoGen, San Diego, CA, USA), ADU-S100 (MCE, Monmouth, NJ, USA), MSA-2 (MCE), E7766 (Eisai Inc.), DMXAA (InvivoGen), brefeldin A (Sigma Aldrich) and H-151 (Tocris, Minneapolis, MN, USA). HT-DNA, cGAMP, and E7766 were dissolved in H_2_O, whereas all other reagents were dissolved in DMSO (Fisher, Hampton, NH, USA), all stored at −20 °C.

### 2.2. Quantitative Real Time-PCR Transcriptional Analysis

RNA was extracted from treated cells as indicated in the figure legends using the TRIzol reagent (Ambion, Austin, TX, USA) following the manufacturer’s instructions. cDNA was synthesized with iScript cDNA synthesis kit (Bio-Rad, Hercules, CA, USA) and analyzed using a Bio-Rad CFX qRT-PCR using iTaq Universal SYBR Green Supermix (Bio-Rad). The mRNA fold change was calculated using the 2^−ΔΔCt^ method where GAPDH was used as the housekeeping gene and transcriptional data are shown as relative comparison to the respective vehicle control. Error bars indicate the range from two independent experiments or SEM from 3 or more individual experiments, each performed as technical duplicates. All primers were ordered from Sigma Aldrich and their sequences are forward human *GAPDH*: GCAAATTCCATGGCACCGT, reverse human *GAPDH*: TCGCCCCACTTGATTTTGG, forward human *IFNβ*: CAAGTGTCTCCTCCAAATTGCTCTC, reverse human *IFNβ*: TCTCCTCAGGGATGTCAAAGTTCAT, forward human *IFIT1*: CTGCCTATCGCCTGGATGGCTTTAA, reverse human *IFIT1*: CTGTGAGGACATGTTGGCTAGAGCT, forward human *mt-COX-1*: ATGACCCACCAATCACATGC, reverse human *mt-COX-1*: ATCACATGGCTAGGCCGGAG, forward mouse *Gapdh*: TTCACCACCATGGAGAAGGC, reverse mouse *Gapdh*: GGCATGGACTGTGGTCATGA, forward mouse *Ifnβ*: CTGCGTTCCTGCTGTGCTTCTCCA and reverse mouse *Ifnβ*: TTCTCCGTCATCTCCATAGGGATC. Data were subjected to a one-way ANOVA or two-way ANOVA for statistical analysis as described in figure legends.

### 2.3. Immunofluorescence

HCC1937 cells were plated on glass coverslips in a 24-well plate at approximately 100,000 cells per well. After overnight incubation, the cells were treated with the indicated concentrations of drugs for the indicated time. Cells were then fixed with 4% paraformaldehyde in PBS for 20 min at RT, permeabilized, and blocked at RT for 90 min with 0.5% Triton X-100 plus 10% bovine calf serum in PBS before immunostaining. The primary and secondary antibodies were diluted in PBS containing 0.5% Triton X-100 and 1% BSA, incubated overnight at 4 °C or 90 min at room temperature respectively. Cell nuclei were stained with DAPI for 20 min, followed by imaging using a Nikon Eclipse 80i fluorescence microscope using a 100 × objective lens. The primary antibodies used for immunofluorescence studies include β-tubulin (Ab6046, Abcam, 1:1000) and GBF1 (12116, BD Transduction Laboratories). The secondary antibodies used for immunofluorescence studies include Mouse IgG Alexa 594 (A11032, Invitrogen, 1:2000) and Rabbit IgG Alexa 488 (A21206, Invitrogen, 1:2000) All images were focused stacks using NIS elements software (Melville, NY, USA).

### 2.4. Immunoblotting

Cell lysates were prepared in Cell Extraction Buffer (Invitrogen) supplemented with protease inhibitor cocktail (Sigma-Aldrich, St Louis, MO, USA), 1 mM PMSF, and 200 µM sodium orthovanadate. Approximately 10~15 ug of whole-cell lysate was resolved by SDS PAGE using Bolt 10% Bis-Tris Plus gels (Thermo Fisher Scientific, Waltham, MA, USA) and transferred to Immobilon-FL PVDF membranes (EMD Millipore) for immunodetection of proteins. After 90 min blocking at RT in LI-COR blocking buffer, the blots were incubated overnight at 4 °C with primary antibodies and 90 min at RT with mouse or rabbit IRDye^®^ secondary antibodies (LI-COR Biosciences, Lincoln, NE, USA). Blots were then imaged on an Odyssey FC (LI-COR). Immunoblot images were analyzed by Image Studio (Lite Ver 5.2). The primary antibodies used for immunoblotting include anti-TBK1 (3504, Cell Signaling Technology, 1:1000), anti-p-TBK1 (5483, Cell Signaling Technology, 1:1000), anti-IRF3 (Sc-33641 Santa Cruz, 1:500), anti-p-IRF3 (29047, Cell Signaling Technology, 1:1000), anti-STING (13647, Cell Signaling Technology, 1:500), anti-p-STING (50907, Cell Signaling Technology, 1:1000), and anti-GAPDH (97166, Cell Signaling Technology, 1:1000).

### 2.5. Intracellular IFNβ Flow Cytometry

THP-1 cells were treated with 1 µL/mL GolgiPlug (555029, BD Biosciences, Franklin Lakes, NJ, USA) with 100 nM eribulin or the DMSO vehicle control with or without 10 μM ADU-S100 for 6 h or 33 μM MSA-2 for 3 h. Cells were then collected, blocked with TruStain FcX block (BioLegend, San Diego, CA, USA, 1:100), and stained with Zombie NIR viability stain (BioLegend, 1:400) for 15 min in the dark. After fixation and permeabilization, cells were incubated with anti-IFNβ FITC (PBL Assay Science, Piscataway, NJ, USA, 1:50) for 20 min in the dark. Intracellular IFNβ staining in live cells was determined using the BD FACS Celesta at the UTHSCSA Flow Cytometry Facility and analyzed with FlowJo software (BD Biosciences).

### 2.6. IFNβ ELISA

THP-1 cells were seeded in either 24-well or 96-well plates at a density ranging from 2 × 10^5^ to 2 × 10^6^ cells/mL. The cells were treated with indicated concentrations and times in figure legends. For the synergism experiments, concentrations ranging from 5 to 1000 nM eribulin were used with a DMSO vehicle control indicated as 0 nM. The concentrations for ADU-S100 ranged from 2 to 40 μM with a DMSO vehicle control indicated as 0 μM. The concentrations for E7766 ranged from 0.1 to 5 μM with a sterile H_2_O vehicle control. Poly I:C treatment was done by transfecting THP-1 cells with poly I:C (InvivoGen, San Diego, CA, USA) using the Lipofectamine 3000 (Thermo Fisher Scientific) reagent. To determine IFNβ levels by ELISA, the cell media was harvested after treatment by centrifuging the cells at 500× *g* for 5 min. 50 μL of sample was used in the High Sensitivity Human IFN Beta ELISA Kit (PBL Assay Science, Piscataway, NJ, USA). The ELISA was completed following the instructions provided in the kit. The final plate absorbances were read at 450 nm using the SpectraMax microplate reader (Molecular Devices, San Jose, CA, USA). Final concentrations were calculated by using the standard curve included in developing the ELISA plate. Drug combination responses to evaluate synergy using the final IFNβ concentrations were calculated based on a ZIP reference model using SynergyFinder [30].

### 2.7. In Vivo Immune Studies

For the analysis of immune populations in the 4T1 tumor model, six-week-old female BALB/c mice were purchased from Envigo (Indianapolis, IN, USA). Tumors were established by injecting 50,000 4T1 cells in PBS bilaterally in the flank. Mice were monitored until tumors reached an average volume of 100 mm^3^ when they were assigned to one of four groups: eribulin + tumor (*n* = 4 mice), paclitaxel + tumor (*n* = 3 mice), no treatment + tumor (*n* = 3 mice), and no tumor + no treatment (*n* = 3 mice). Dosing of 1.0 mg/kg eribulin Q7Dx2 or 20 mg/kg paclitaxel Q2Dx6 was administered intraperitoneally for two weeks. Tumor volumes and weight were monitored throughout the dosing timeline. The spleen and inguinal draining lymph nodes were collected at the end of dosing and processed for flow cytometric analysis. Tissues were processed and filtered through 70 μm filters (CellTreat, Pepperell, MA, USA) to produce single-cell suspensions. Spleen samples were treated with 1–2 mL of ACK lysing buffer (Thermo Fisher Scientific) for 7 min to remove red blood cells. Cell counts of each sample were collected to stain 5 million cells per sample. Samples were blocked with TruStain FcX PLUS (BioLegend, 1:100) and Zombie NIR Viability stain (BioLegend, 1:400) for 15 min. Then, the samples were stained with CD45 Alexa Fluor 700 (1:400), CD45R PerCP-eFluor710 (1:200), CD19 Pacific Blue (1:400), CD3 Brilliant Violet 510 (1:200), CD8a PerCP/Cy5.5 (1:200), CD4 Alexa 488 (1:400), CD62L Brilliant Violet 711 (1:400), and CD44 PE/Cy7 (1:200), all obtained from BioLegend. After staining, cells were analyzed using the Cytek Aurora (Cytek, Fremont, CA, USA) at the UTHSCSA Flow Cytometry Shared Resource Facility and further analyzed using FlowJo (BD Biosciences). Our gating strategy to identify and quantify activated effector T-cells under each treatment condition started with gating for the lymphocyte population in the forward and side scatter flowplot, followed by further gating for single cells, after which live cells were identified using a Zombie NIR viability dye. We then successively gated for CD45^+^, CD3^+^, CD19^−^, and CD45R^−^ to identify T-cells that were then further classified as either CD4^+^ or CD8^+^ T-cells. Each population of CD4^+^ and CD8^+^ T-cells was then analyzed for CD44 and CD62L expression to identify respective naïve (CD62L^+^ CD44^−^) and effector (CD62L^−^ CD44^+/−^) cells. For an immune analysis of non-tumored animals, 5-week-old FVB/N noncarrier mice were purchased from the Jackson Laboratory (Bar Harbor, ME, USA). Mice were randomly assigned to four groups, including eribulin + ADU-S100, eribulin, ADU-S100, and vehicle (*n* = 3 mice). Animals were dosed with 0.7 mg/kg eribulin or vehicle i.p. in 200 µL PBS Q4Dx5 and with 50 μg ADU-S100 or vehicle Q4Dx3 s.c. Weight was monitored throughout the dosing timeline. Four days after the last dose of eribulin, the spleens were collected from the mice and processed, stained, and analyzed in the same protocol outlined previously for the 4T1 BALB/c model.

### 2.8. Antitumor Trial

Five-week-old female FVB/N-Tg (MMTV-PyVT) mice were purchased from the Jackson Laboratory. Mice were monitored until tumors began to grow along the mammary fat pad. Mice were assigned to cohorts to begin treatment once their largest tumors were 100–800 mm^3^. Due to the multiple tumors and large variability in tumor sizes in this spontaneous tumor model, mice were assigned to different groups to include a wide range of starting tumor sizes. Each group was assigned at least one mouse with a tumor starting at 500–800 mm^3^ on day 1. The rest of the mice were pair-matched to groups based on their average tumor sizes and total tumor burden. Four groups included eribulin + ADU-S100 (*n* = 4 mice, 58 tumors), eribulin (*n* = 2 mice, 20 tumors), ADU-S100 (*n* = 2 mice, 29 tumors), and vehicle (*n* = 2 mice, 28 tumors). Systemic injections of 0.7 mg/kg eribulin or vehicle in 200 µL PBS were given intraperitoneally Q4Dx5. Local tumor injections of 50 µg ADU-S100 or vehicle in 50 µL PBS were given Q4Dx2 into the largest tumor on each mouse. Tumor volume was measured by calipers (length × width × height) every four days, and weight was monitored throughout the trial until the largest tumor of the individual mouse reached the endpoint tumor volume of 1500–2000 mm^3^ in accordance with our IACUC protocol. Statistical significance was determined by two-way ANOVA with Tukey’s posthoc test.

## 3. Results

### 3.1. Eribulin Enhances the IFNβ Response of STING Agonists in Immune Cells

To investigate the interaction between eribulin and STING agonists, we first determined the time-dependent expression of IFNβ in the human monocytic leukemia cell line THP-1 after transfection with HT-DNA as a mechanism to increase cytosolic DNA and promote STING activation. This response increased over time with strong induction observed 24 h after DNA transfection (Figure 1A). When transfected THP-1 cells were co-treated with 100 nM eribulin, a clinically relevant concentration [31], the cells expressed significantly higher levels of IFNβ after 24 h of treatment than with HT-DNA alone (Figure 1A). The ongoing development of STING agonists for cancer treatment prompted a comprehensive analysis of the ability of eribulin to enhance the activity of cyclic di-nucleotide STING agonists (e.g., cGAMP, ADU-S100, E7766), as well as small molecule non-cyclic di-nucleotide STING agonists (e.g., MSA-2) (Figure 1B) [32]. All STING agonists evaluated induced the concentration and time-dependent expression of IFNβ in THP-1 cells (Appendix A and Figure 1C–F) and eribulin was able to enhance the IFNβ response mediated by a sub-maximal concentration of each agonist at the timepoint when maximal expression was observed (Figure 1C–F). The ability of eribulin to enhance the IFNβ expression by STING agonists was also concentration-dependent and detectable at eribulin concentrations as low as 50 nM (Appendix A). The transient expression of IFNβ in response to STING ligands is consistent with the literature [33,34] and allows for initiation of a downstream immunological response while avoiding toxicity associated with continuous interferon expression [35,36]. No evidence of cytotoxicity was observed in cells at the concentrations and timepoints evaluated consistent with previous reports [6,23,25,27]. To determine if eribulin’s mechanism of enhancement required the full 6 h co-incubation with ADU-S100, we added eribulin at staggered times during the 6 h time course. We found that the addition of eribulin for as little as the final 2 h of the 6 h ADU-S100 treatment was sufficient to increase IFNβ expression (Appendix A). This demonstrates that although the maximal effect of ADU-S100 as a single agent requires 6 h (Figure 1D), eribulin-mediated enhancement is not required for the first 4 h, supporting a relatively acute role of microtubule destabilization on the enhancement of STING-dependent IFNβ induction. To interrogate if eribulin-enhanced IFNβ expression was dependent on the STING signaling pathway, THP-1 cells were pretreated with the STING antagonist H-151, followed by treatment with the STING agonists and eribulin. In all cases, the STING inhibitor attenuated the IFNβ expression induced by eribulin in combination with STING agonists, suggesting this enhancement is mediated by canonical STING signaling (Figure 1G).

The ability of eribulin to enhance the transcription of IFNβ by STING agonists prompted a downstream analysis of IFNβ protein levels and functional interferon response. THP-1 cells were treated with eribulin and STING agonists at times and concentrations based on the transcriptional response (Figure 1 and Appendix A) and secretion of IFNβ measured by ELISA. Consistent with the transcriptional induction, eribulin robustly enhanced the secretion of the IFNβ cytokine from THP-1 cells when combined with the STING agonists ADU-S100 (Figure 2A), MSA-2 (Figure 2B), or E7766 (Figure 2C). This eribulin-mediated increase in IFNβ production occurred in live THP-1 cells when combined with STING agonists as determined by intracellular flow cytometry when cytokine release was inhibited by the GolgiPlug inhibitor and cells were stained with a viability dye (Figure 2D,E). When secreted, type I interferons, including IFNβ, subsequently initiate a signaling cascade that leads to the transcriptional expression of hundreds of interferon-stimulated genes (ISGs) [37]. To test the downstream effects on ISG expression 

When eribulin is combined with STING agonists, we measured the transcriptional induction of the ISG IFIT1 in THP-1 cells treated with ADU-S100 and eribulin. Combination treatment of THP-1 cells with eribulin and ADU-S100 led to an enhanced transcriptional expression of IFIT1 consistent with the higher secretion of IFNβ protein (Figure 2F). The ability of eribulin to enhance IFNβ expression by STING agonists was extended to murine immune cells where we could evaluate the interaction with the murine-specific STING ligand DMXAA [38]. Indeed, eribulin was able to enhance the IFNβ response to DMXAA in murine RAW 264.7 macrophage cells similar to the effects observed in human THP-1 cells with other STING agonists (Figure 2G). Together, our data demonstrate that eribulin can enhance IFNβ expression initiated by a diverse array of STING agonists at both the transcriptional and protein levels in both human and murine immune cells.

### 3.2. Eribulin Synergizes with STING Agonists

The ability of eribulin to promote a robust increase in IFNβ expression when combined with STING agonists invoked the hypothesis that this interaction was synergistic. To test this possibility, a series of concentrations of STING agonists (ADU-S100 or E7766) and eribulin were added to THP-1 cells as single agents or in combination, and the secretion of IFNβ was monitored by ELISA. Cell density was reduced as compared to previous ELISAs (Figure 2) in order to allow for detection of synergistic effects in the linear range of the assay, owing to differences in the relative potency of STING agonists for detectable IFNβ induction in these experiments. Consistent with our previous studies [6], the low induction of the STING pathway by eribulin alone did not yield any detectable increase in secreted IFNβ up to a concentration of 1 µM, but the addition of eribulin at concentrations as low as 5 nM was sufficient to increase IFNβ secretion induced by the STING ligands (Figure 3A,C). To determine if the effects of eribulin and STING agonists were synergistic, we utilized zero interaction potency (ZIP) analysis that combines the advantages of the Loewe and Bliss synergism models while reducing the false positive rate [39]. ZIP synergy analysis performed using SynergyFinder [30] found that the combination of eribulin with ADU-S100 yielded a synergy score of 36.6 and the combination of eribulin with E7766 yielded a synergy score of 16.7 (Figure 3B,D). These values demonstrate profound synergy between eribulin and pharmacological STING agonists for IFNβ production, surpassing the accepted synergism score thresholds of 5–10 reported in the literature [39,40]. Collectively, these findings demonstrate that the combination of eribulin with STING agonists leads to a synergistic increase in IFNβ secretion.

### 3.3. Eribulin Enhances the IFNβ Response of STING Agonists in TNBC Cells

Eribulin is used for the treatment of advanced breast cancer, including TNBC [41,42]. Since STING expression is heterogenous in TNBC cancer cells and tumors [6,43], this prompted the hypothesis that eribulin could augment the IFNβ response initiated by STING agonists in TNBC cells with a functional STING pathway. We selected three TNBC cell lines, two that express STING (HCC1937 and HCC1806) and the MDA-MB-453 line that does not express detectable STING protein (Figure 4A). As expected, the STING-expressing HCC1937 and HCC1806 cells, but not the STING-deficient MDA-MB-453 cells, responded to HT-DNA challenge with increased IFNβ expression as compared to vehicle controls (Figure 4B). Similar to the effects in THP-1 cells, eribulin was able to significantly enhance the IFNβ response induced by HT-DNA in the STING-expressing TNBC cells but did not enhance this response in the STING-deficient MDA-MB-453 cells (Figure 4B). In contrast to the shared response of HCC1937 and HCC1806 cells to HT-DNA, the pharmacological STING agonists had differential efficacy in these lines. The HCC1937 cells responded to MSA-2 but not to ADU-S100 or E7766, whereas HCC1806 cells responded to ADU-S100 and E7766, but not MSA-2 (Figure 4C–E and Appendix A). While outside the scope of this study, these data suggest that cancer cell lines can have specificity in their response to pharmacological STING agonists in spite of their shared response to HT-DNA. This is likely due to the known differences between these compounds, including the fact that the cyclic dinucleotides E7766 and ADU-S100 suffer from poor membrane permeability and stability [44], whereas MSA-2 has improved bioavailability but requires dimerization in a pH-dependent manner to serve as a STNG ligand [45]. Regardless of the differential response of TNBC cancer cells to these pharmacologically diverse STING agonists, eribulin was able to enhance the IFNβ response in all instances where the STING agonist produced a response on its own. These results support our overarching premise that eribulin can enhance IFNβ expression in response to diverse mechanisms of STING agonism in both immune and TNBC cells.

### 3.4. Enhancement of STING Signaling Occurs Downstream of Microtubule Depolymerization and Is Not Shared with Microtubule Stabilizing Taxanes

MTAs are subclassified into two groups based on their net effect on microtubule structures; microtubule-stabilizing agents, which include the taxanes and microtubule-destabilizing agents, such as eribulin, vinorelbine, and combretastatin A-4 [46]. Microtubule destabilizers can be further stratified depending on their binding location on the αβ-tubulin heterodimer. For example, eribulin and vinorelbine bind within the vinca domain, whereas combretastatin A-4 binds within the colchicine site [46]. To determine whether the microtubule depolymerization induced by eribulin is the mechanism of action that leads to the enhancement of IFNβ expression by STING agonists, we compared the effects of eribulin to other microtubule stabilizers and destabilizers. As expected, treatment of HCC1937 TNBC cells with 100 nM of the destabilizers eribulin, vinorelbine, or combretastatin A-4 led to the complete depolymerization of cellular microtubules, whereas treatment with 100 or 500 nM paclitaxel led to microtubule bundling (Figure 5A). We found that the destabilizers eribulin, vinorelbine, and combretastatin A-4, but not the stabilizer paclitaxel, enhanced the transcription and secretion of IFNβ in combination with ADU-S100 or cGAMP in THP-1 cells (Figure 5B–D). This same trend was observed in HCC1937 TNBC cells in combination with MSA-2 (Figure 5E). The finding that all three microtubule destabilizers that bind within two different tubulin binding sites, but not paclitaxel, can enhance the IFNβ response by STING agonists in both immune and TNBC cells suggests that the mechanism of action of eribulin-mediated enhancement of STING-dependent IFNβ expression is a result of its on-target microtubule depolymerizing activity regardless of binding site. These data also demonstrate that the ability of eribulin to enhance STING agonist activity is not due to the antiproliferative or cytotoxic effects shared with the taxane class of microtubule stabilizers, providing an important mechanistic distinction between these two classes of microtubule targeted chemotherapeutics.

### 3.5. Eribulin Enhances STING-Dependent IFNβ Expression through TBK1-IRF3 Signaling

The STING-dependent induction of interferon signaling requires STING exit from the endoplasmic reticulum (ER). This can occur upon the binding of an endogenous or pharmacological ligand which induces a conformation change in STING that allows for trafficking through the secretory pathway [47]. After leaving the ER in response to ligand binding, STING recruits the kinase TBK1 which subsequently phosphorylates itself, STING, and the transcription factor IRF3 within the ER–Golgi intermediate compartment (ERGIC)/cis-Golgi before STING is degraded in the lysosome (Figure 6A) [48]. We have previously shown that eribulin as a single agent can activate STING-dependent IFNβ expression downstream of the release of mitochondrial DNA (mtDNA) into the cytoplasm [6]. However, this induction is much lower than endogenous or pharmacological STING ligands and is not sufficient to detect increased IFNβ secretion by ELISA. To interrogate whether the release of mtDNA by eribulin was necessary for enhancing the IFNβ response mediated by STING agonists, we grew THP-1 cells in ethidium bromide to deplete them of mitochondrial DNA (Rho^0^ THP-1) (Appendix A). ADU-S100-induced interferon secretion was enhanced by eribulin in these Rho^0^ THP-1 cells, suggesting that the eribulin-mediated enhancement of STING agonist activity was distinct from its ability to promote mtDNA release into the cytoplasm (Figure 6B). This finding led us to empirically test other mechanisms by which eribulin could enhance the activity of STING ligands, including the regulated trafficking of STING from the ER, the phosphorylation of TBK1/IRF3/STING, and lysosomal degradation of STING. To test if the known disruption of intracellular organelles downstream of microtubule destabilization [49,50,51] could override controlled ER exit, we pre-treated THP-1 cells with brefeldin A (BFA) at a concentration that caused complete disruption of the cis-Golgi (Appendix A), followed by treatment with the combination of eribulin and ADU-S100. BFA abrogated the induction of IFNβ expression by the STING agonist with or without eribulin, suggesting that eribulin was not overriding the regulated exit of STING from the ER in spite of destabilizer-induced organelle disruption [49,50,51] (Figure 6C). We next examined the phosphorylation of TBK1, IRF3 and STING over time after treating THP-1 cells with eribulin and ADU-S100 where we observed that eribulin promoted an increase in phosphorylated IRF3 and STING as compared to ADU-S100 alone, particularly at the 6 h timepoint when the effects on IFNβ expression are most pronounced (Figure 6D–E). Finally, we also tested the hypothesis that the organelle fragmentation caused by microtubule destabilizers [49,50,51] could enhance STING-dependent signaling by inhibiting its lysosomal trafficking and degradation. Similar to previous reports [52], we did not observe significant STING degradation in the THP-1 cells upon activation (Figure 6E and Appendix A). However, in HCC1937 cells, where STING levels were effectively decreased by STING activation with MSA-2, eribulin did not inhibit this decrease (Figure 6F), suggesting that microtubule destabilization is not inhibiting STING degradation in the lysosome.

### 3.6. Eribulin Enhances the Immune Response of Multiple Agonists

Our finding that eribulin promoted increased IRF3 phosphorylation as compared to ADU-S100 alone in THP-1 cells (Figure 6D) invoked the possibility that eribulin could be enhancing TBK1-dependent IRF3 phosphorylation downstream of STING activation. Both the cGAS-STING and RIG-I-MAVS pathways converge on activation of the TBK1/IRF3 transcription factors to initiate the expression of IFNβ in response to DNA and RNA ligands, respectively [48,53]. We hypothesized that if eribulin was enhancing STING-dependent IFNβ expression through downstream activation of TBK1/IRF3, then it would also enhance the IFNβ response generated by RIG-I RNA-based ligands, such as poly I:C. To test this, we treated THP-1 cells with combinations of poly I:C and eribulin and monitored the secretion of IFNβ by ELISA. Indeed, eribulin enhanced IFNβ secretion induced by poly I:C over a range of poly I:C concentrations (Figure 7A), supporting the hypothesis that eribulin can enhance IFNβ expression in response to the activation of multiple innate immune sensing pathways that converge on activation of the TBK1/IRF3 transcription factors. The ability of eribulin to function as an agonist of multiple innate immune sensing pathways prompted us to test the hypothesis that it could enhance the in vivo immunogenic response to a tumor independent of direct effects on tumor growth. For this study, we utilized the 4T1 BALB/c syngeneic murine TNBC tumor model since 4T1 cells are highly resistant to the antiproliferative and cytotoxic effects of microtubule targeted drugs [54]. We focused on identifying whether CD4^+^ or CD8^+^ T-cell populations were shifting from naïve to effector cells using loss of CD62L as a marker of activation. We found that BALB/c mice bearing 4T1 tumors had a significant increase in the activation status of CD4^+^ T-cells in their spleen as compared to non-tumored animals as measured by the loss of CD62L in these T-cells, (Figure 7B). This immunogenic response to the tumor was further enhanced by eribulin, but not by paclitaxel, as observed by the increase in activated CD4^+^ T-cells in both the spleen and the tumor draining lymph nodes of eribulin-treated tumored animals (Figure 7B,C). We did not observe a significant difference in the activation of CD8^+^ cells or the total population of CD4^+^ or CD8^+^ T-cells or B-cells. The eribulin-mediated increase in T-cell activation was determined to be tumor-specific as it was not observed in non-tumored animals treated with eribulin alone (unpublished observation). The fact that we observed increased T-cell activation in this model, which is resistant to the antitumor effects of eribulin due to P-glycoprotein expression [55] such that tumors grew at the same rate regardless of drug treatment, demonstrates that the eribulin-mediated activation of CD4^+^ T-cells is dependent on the presence of a tumor, but independent of direct cytotoxicity against the tumor. Furthermore, neither eribulin nor the STING agonist ADU-S100 alone was able to promote a significant increase in activated CD4^+^ T-cells in the spleens of non-tumored FVB/N mice, but an increase in this population was observed when these drugs were combined (Figure 7D). Together, these in vivo data support our in vitro observations that eribulin can effectively function as an immune adjuvant, enhancing the immunological response to either a STING agonist or tumor.

### 3.7. Eribulin Shows Improved Antitumor Efficacy in Combination with ADU-S100

The ability of eribulin to synergize with STING agonists in vitro (Figure 3) and enhance the in vivo immunological response to either tumors or pharmacological STING agonists (Figure 7) prompted an evaluation of the antitumor efficacy of eribulin with ADU-S100 in an aggressive genetically induced mammary tumor model. For this study, we utilized the spontaneous murine mammary tumor model FVB/NJ MMTV-PyVT, which is the strain background where we observed a significant increase in activated T-cells with this drug combination (Figure 7D). This transgenic tumor model provides unique opportunities and challenges as it allows for the evaluation of sustained immunological responses with constant rechallenge from spontaneous tumor growth with individual animals possessing multiple tumors. Animals were enrolled into cohorts pair-matched by tumor size when their largest tumors were between 200–800 mm^3^ and treated with eribulin alone, ADU-S100 alone, the combination, or vehicle controls. Mice were dosed with 0.7 mg/kg eribulin or eribulin vehicle by intraperitoneal (i.p.) injection Q4Dx5. ADU-S100 was given by intratumoral injection of 50 µg into the largest tumor on days 1 and 5 [26]. Individual tumor volumes were monitored, and animals were euthanized when the largest tumor reached 1500 mm^3^ (Figure 8A). At the end of the 21-day treatment period, significant antitumor efficacy was observed for the combination of ADU-S100 and eribulin, but not for either treatment alone, as determined by measuring all tumors that grew over the treatment period (Figure 8B,C) or only measuring tumors that were established at the initiation of dosing (Figure 8D). These data demonstrate that the combination of eribulin with the STING agonist ADU-S100 provides antitumor efficacy against established tumors, as well as on tumors that spontaneously appeared during treatment. It is important to note that the combination also promoted tumor regression in several animals (Figure 8C). Although the mice were euthanized at different times due to reaching tumor endpoint criteria, we continued making tumor measurements in animals that had not reached endpoint for up to 24 days after dosing was halted, and individual tumor volumes over the duration of the trial are represented as spider plots (Figure 8E). This visualization demonstrates that at least one mouse with a larger tumor (500–800 mm^3^) was assigned to each non-vehicle group. Notably, these larger tumors in the groups receiving eribulin or ADU-S100 alone rapidly reached endpoint tumor volume and many of the smaller tumors continued to grow even during the treatment period. In contrast, the largest tumor in the mice receiving the combination treatment decreased in tumor volume over the duration of treatment and each of the smaller tumors also regressed during treatment, providing an overall survival advantage of these animals even after dosing was halted. Overall, the combination of eribulin with ADU-S100 has improved antitumor efficacy over either agent alone as determined by the inhibition of tumor growth during treatment, as well as a potential overall survival advantage.

## 4. Discussion

This study builds on our previous work showing that the microtubule destabilizer eribulin can activate the cGAS-STING pathway by mediating the release of mitochondrial DNA as a downstream consequence of microtubule disruption [6]. While eribulin as a single agent was sufficient to promote a significant increase in STING-dependent interferon production as measured by qRT-PCR and intracellular flow cytometry, this induction was not sufficient to result in a detectable increase in IFNβ by ELISA [6]. In the current study, we hypothesized that eribulin could enhance the production of IFNβ in combination with STING agonists to a greater extent than either drug alone. Indeed, our data demonstrate the ability of eribulin to enhance STING-dependent interferon expression induced by structurally diverse STING agonists, including those currently undergoing clinical evaluations, in both immune and TNBC cells. The profound level of synergism resulting from this combination treatment, combined with the ability of eribulin to enhance STING and RIG-I/MDA-5 agonists in vitro and in vivo, provided the rationale for evaluations of these agents in a recurrent murine model of breast cancer where the combination of eribulin and a STING agonist showed improved antitumor efficacy compared to either agent alone.

Our findings are timely given the active clinical development of STING agonists as a promising new class of cancer immunotherapeutics [24]. Activation of the cGAS-STING immune sensing pathway is a validated strategy to increase tumor immunogenicity and improve antitumor efficacy in preclinical and clinical studies [56,57]. Natural STING ligands, including cGAMP, provide proof of principle that STING activation can promote antitumor efficacy, but these ligands suffer from poor cell penetration and metabolic liabilities [11,58,59] that necessitate intratumoral injections and are not maximally effective against all of the common variants of human STING [60,61]. Diverse pharmacological STING agonists have been developed with improved physiochemical properties and the ability to interact with major human STING variants. These include the cyclic dinucleotides ADU-S100 and E7766, as well as the non-nucleotide small molecules MSA-2 and SNX281. While it is unclear which of these agents may provide the greatest clinical benefit in ongoing clinical trials, we demonstrate that eribulin significantly enhances interferon production when combined with each of these chemically distinct classes of STING agonists. These data suggest that the use of eribulin or potentially other microtubule depolymerizers in combination with STING agonists could provide a dual role of enhancing interferon production while also serving as an effective chemotherapy backbone.

The mechanism of eribulin-dependent enhancement of STING signaling reported here is downstream of its microtubule destabilizing activity since this phenotype is shared with structurally diverse destabilizers, including vinorelbine and combretastatin A-4, but not with the microtubule stabilizer paclitaxel. While microtubule stabilizing and destabilizing chemotherapeutics lead to shared antimitotic effects after prolonged treatment, they have different effects on interphase microtubule structures that are observed within 2 h of treatment [62]. Our finding that eribulin enhances STING agonist induced IFNβ expression within 3–6 h in a manner distinct from paclitaxel indicates that this acute effect on STING signaling is separate from the antimitotic effects of eribulin. However, it is important to note that other studies have demonstrated that chronic treatment with either stabilizers or destabilizers can activate the cGAS-STING pathway after prolonged mitotic arrest due to chromosome instability and micronucleation that can result from mitotic slippage [63]. These micronuclei have weak membranes that can release double-stranded DNA into the cytoplasm and activate the cGAS-STING pathway, leading to IFNβ expression when cells are exposed to either microtubule stabilizing or destabilizing MTAs for 24 h or longer [63]. We propose that the acute, interphase effects that are unique to the microtubule destabilizing MTAs would be particularly beneficial in solid tumors that have a low mitotic index [64] and could enhance STING signaling within the timeframe that they are present in serum at the concentrations where we observe these effects [31].

The mechanism by which microtubule depolymerization enhances STING signaling is independent of the mitochondrial DNA release we previously reported for eribulin [6], suggesting an additional mechanism. Since canonical STING signaling requires ligand-dependent trafficking from the ER to the ERGIC/Golgi and it is well established that microtubule disruption promotes collapse of the ER, fragmentation of the Golgi, and disruption of other organelles [49,50,51], we interrogated whether eribulin could override the requirement for regulated ER exit. The ability of brefeldin A, an inhibitor of ER to Golgi trafficking, to completely abrogate the ability of eribulin and to enhance STING signaling demonstrates that eribulin was not sufficient to override this regulated delivery of STING to the ERGIC compartment where it can recruit and phosphorylate TBK1 and IRF3. Additionally, we did not observe any evidence that STING degradation was inhibited by decreased trafficking to the lysosome for degradation. The most significant impact of eribulin on the STING signaling pathway was a dramatic increase in the phosphorylation of IRF3 when THP-1 cells were treated with the combination of eribulin and ADU-S100, indicating that the enhanced IFNβ expression could be downstream of STING at the level of TBK1/IRF3. Our finding that eribulin also significantly enhanced IFNβ expression induced through the RNA-sensing pathway further supports the role of TBK1/IRF3 activation in the ability of eribulin to synergize with STING agonists. Further studies will interrogate how microtubule depolymerization could increase TBK1/IRF3 phosphorylation in response to activation of either RNA or DNA sensing pathways. However, there is precedence for the ability of microtubule depolymerization to increase TBK1/IRF3 activation through the release of the guanine nucleotide exchange factor-H1 (GEF-H1) from microtubules which interacts with TBK1-IKKε to induce IRF3 phosphorylation [65]. Other transcription factors that mediate immune sensing pathways, including IRF5, NF-κB, and c-Jun [65] also require GEF-H1 for activation, suggesting microtubule destabilizers could enhance diverse pathways involved in the activation of macrophages, dendritic cells, and T-cells. This is consistent with our observations that eribulin enhances the activation of CD4^+^ T-cells in vivo when combined with either a tumor or STING agonist. This pathway could also underlie the previously reported increase in immune infiltration of tumors in patients treated with eribulin [7] and could explain the survival advantage that was reported in the EMBRACE clinical trial of eribulin as compared to physician’s choice in heavily pretreated metastatic breast cancer patients. [66]

We propose that TNBC is the optimal cancer type to evaluate the effects of combining eribulin with STING agonists. Eribulin is currently used in TNBC and it is the most immunogenic breast cancer subtype, making it a logical first choice to evaluate the effectiveness of immunotherapies. MTAs, including both eribulin [67] and the taxanes [68], have undergone extensive evaluations in combination with immune checkpoint inhibitors in TNBC and this has already led to the FDA approval of paclitaxel in combination with pembrolizumab. However, an obstacle in the effective use of immune checkpoint inhibitors is the need to increase tumor immunogenicity to bring immune cells into the tumor [69]. One mechanism of action that can effectively promote the transition from an immunologically “cold” tumor microenvironment into an environment that would be receptive to checkpoint inhibition is activation of STING signaling [11]. Indeed, profound antitumor efficacy has been observed with combinations of STING agonists and immune checkpoint inhibitors [23,27,70] and many of the clinical trials evaluating STING agonists are being conducted in combination with checkpoint inhibitors [24]. The finding that eribulin further enhances STING signaling makes it a prime candidate for inclusion as a chemotherapy backbone for this combination therapy, particularly if long-term durable responses are not observed in the clinical trials of STING agonists and immune checkpoint inhibitors on their own.

## 5. Conclusions

MTAs are a widely used and effective class of chemotherapeutics that are underappreciated for their distinct non-mitotic effects. We show that the microtubule destabilizer, eribulin, is unique from microtubule stabilizing taxanes in its ability to function as an enhancer of innate immune signaling pathways, including STING mediated interferon production. These findings could underlie the significant survival advantage observed in patients treated with eribulin and provide an opportunity to enhance the antitumor immunological response using a currently approved chemotherapeutic, particularly in combination with STING agonists and/or immune checkpoint inhibitors.

## Figures and Tables

**Figure 1 cancers-14-05962-f001:**
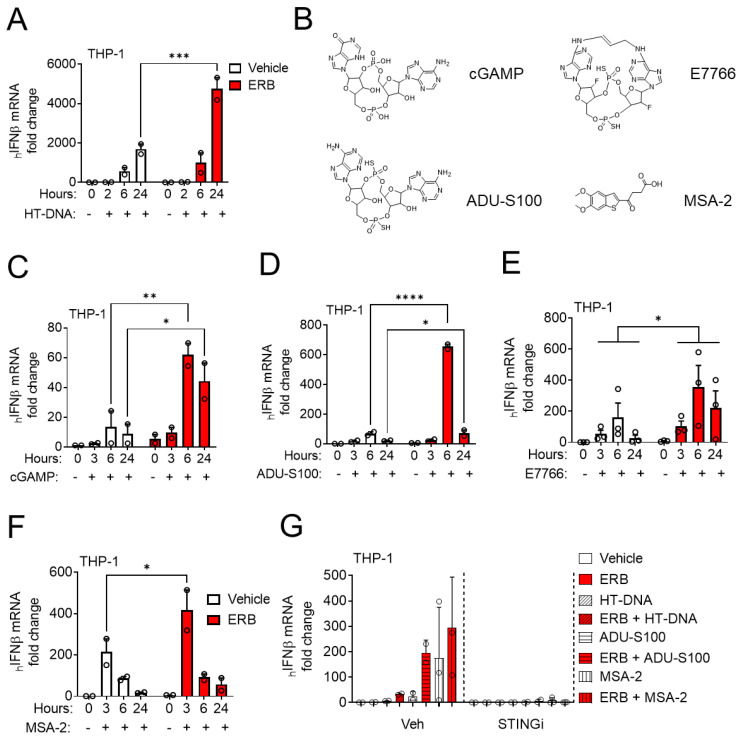
Eribulin enhances *IFNβ* expression initiated by STING agonists. (**A**) Human *IFNβ* mRNA in THP-1 cells transfected with 5 μg/mL HT-DNA and treated with 100 nM eribulin (ERB) or vehicle for 2, 6, or 24 h. (**B**) The chemical structure of the STING agonists examined in this study. Analysis of human *IFNβ* mRNA in THP-1 cells treated with (**C**) 10 μM cGAMP, (**D**) 10 μM ADU-S100, (**E**) 1 μM E7766 or (**F**) 33 μM MSA-2 with 100 nM ERB or vehicle for 3, 6 or 24 h. (**G**) Human *IFNβ* mRNA in THP-1 cells pretreated with 1 µM of the STING inhibitor H-151 for 4 h followed by transfection with 5 μg/mL HT-DNA or treatment with 10 μM ADU-S100 or 33 μM MSA-2 with or without 100 nM ERB. Statistical significance of eribulin as compared to control at each timepoint is shown as determined by a two-way ANOVA (time × eribulin) with Sidak’s posthoc test. * *p* < 0.05, ** *p* < 0.01, *** *p* < 0.001, **** *p* < 0.0001. For panel E, the main effect of eribulin is shown.

**Figure 2 cancers-14-05962-f002:**
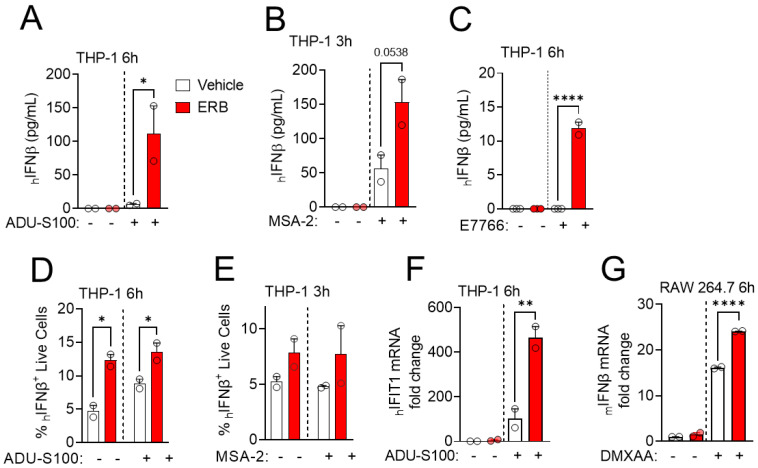
Eribulin stimulates the release of IFNβ and activates downstream interferon stimulated genes. Analysis of human IFNβ secreted from THP-1 cells treated with (**A**) 10 μM ADU-S100 with or without 100 nM eribulin (ERB) for 6 h, (**B**) 33 μM MSA-2 with or without 100 nM eribulin for 3 h, or (**C**) 1 μM E7766 with or without 100 nM eribulin for 6 h. Significance of eribulin on IFNβ secretion was determined by one-way ANOVA with Dunnett’s posthoc test comparing all conditions to the STING agonist alone. Analysis of intracellular human IFNβ protein in THP-1 cells treated with (**D**) 10 μM ADU-S100 with or without 100 nM eribulin for 6 h or (**E**) 33 μM MSA-2 with or without 100 nM eribulin for 3 h. Significance of eribulin on intracellular IFNβ levels was determined by two-way ANOVA with Sidak’s posthoc test. (**F**) Analysis of human *IFIT1* mRNA in THP-1 cells treated with 10 μM ADU-S100 with or without 100 nM eribulin for 6 h. (**G**) Analysis of mouse *IFNβ* mRNA in RAW 264.7 cells treated with 35 μM DMXAA with or without 100 nM eribulin for 6 h. Statistical significance of eribulin on *IFIT1* or *IFNβ* expression compared to STING agonist alone was determined by one-way ANOVA with Dunnett’s posthoc test, comparing all conditions to the combination.* *p* < 0.05, ** *p* < 0.01, **** *p* < 0.0001.

**Figure 3 cancers-14-05962-f003:**
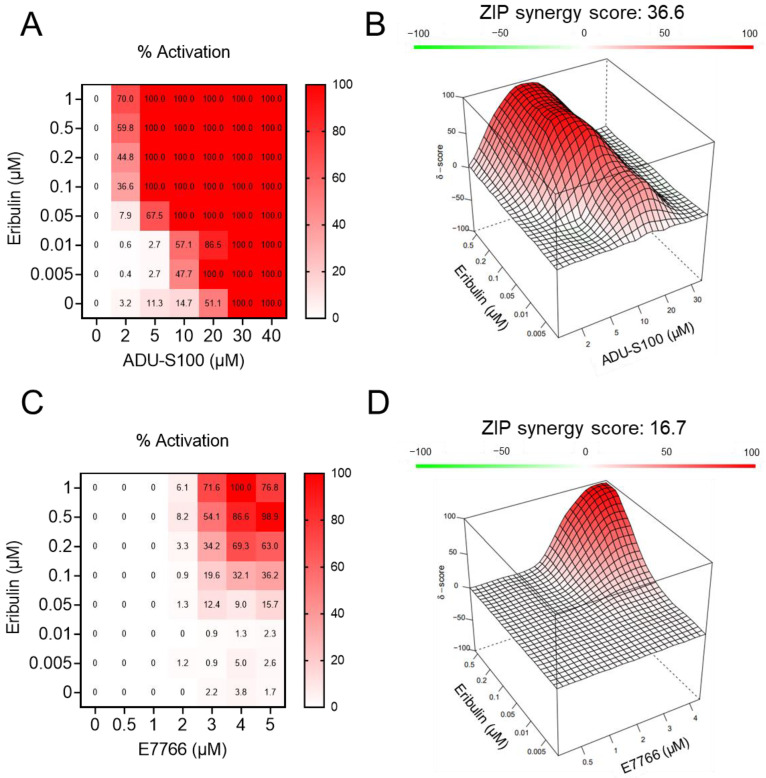
Eribulin synergizes with STING agonists to enhance interferon production. (**A**) Concentration-response matrix of IFNβ release from THP-1 cells treated with indicated combinations of eribulin and ADU-S100 for 6 h as measured by ELISA. (**B**) 3D plot of the delta scores calculated using the ZIP reference model with the resulting ZIP synergy score of the eribulin-ADU-S100 drug interaction. (**C**) Concentration-response matrix of IFNβ release from THP-1 cells treated with indicated combinations of eribulin and E7766 for 6 h as measured by ELISA. (**D**) 3D plot of the delta scores calculated using the ZIP reference model with the resulting ZIP synergy score of the eribulin-E7766 drug interaction. Positive (red) and negative (green) values denote synergy and antagonism, respectively.

**Figure 4 cancers-14-05962-f004:**
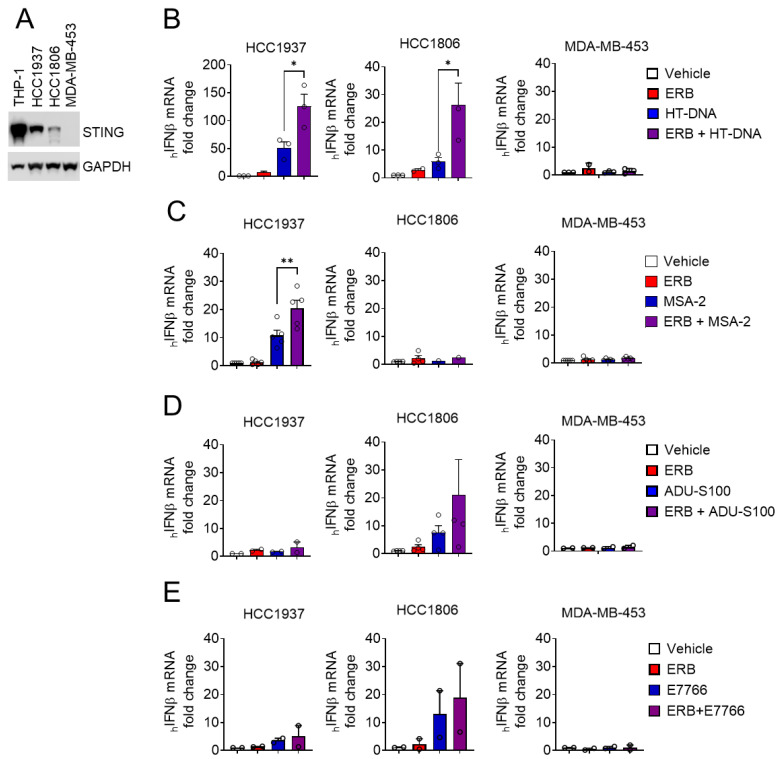
Eribulin enhances *IFNβ* expression induced by STING agonists in TNBC cells. (**A**) Immunoblot analysis of STING and GAPDH in THP-1 and TNBC cells. Human *IFNβ* mRNA in TNBC cell lines (HCC1937, HCC1806, or MDA-MB-453) each transfected with (**B**) 5 μg/mL HT-DNA or treated with (**C**) 33 μM MSA-2, (**D**) 10 μM ADU-S100 or (**E**) 1 µM E7766 with 100 nM eribulin (ERB) or vehicle. Statistical significance of eribulin with the STING agonist as compared to the STING agonist alone is shown as determined by one-way ANOVA with Dunnett’s posthoc test comparing all conditions to the combination. * *p* < 0.05, ** *p* < 0.01. The uncropped Western blots have been shown in Appendix A.

**Figure 5 cancers-14-05962-f005:**
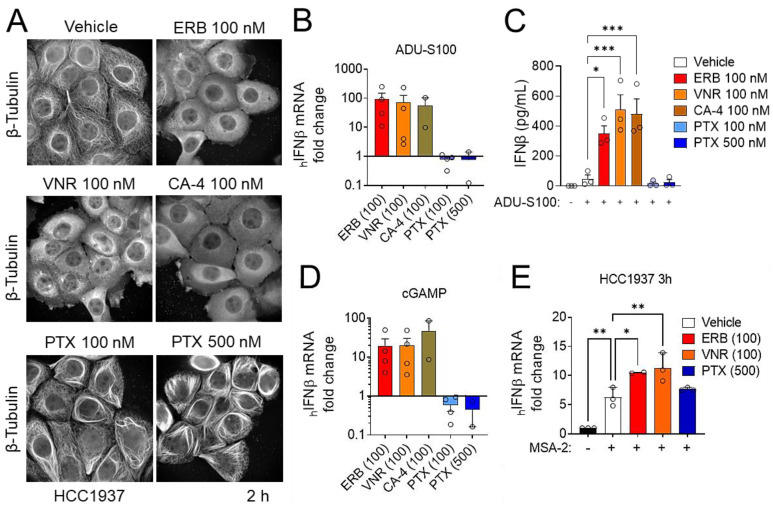
Microtubule destabilizers, but not stabilizers, enhance IFNβ expression induced by STING agonists. (**A**) Immunofluorescence images of microtubules in HCC1937 cells treated with DMSO (vehicle), 100 nM eribulin (ERB), 100 nM vinorelbine (VNR), 100 nM combretastatin A-4 (CA-4), or 100 or 500 nM paclitaxel (PTX) for 2 h. (**B**) Analysis of human *IFNβ* mRNA in THP-1 cells treated with 10 μM ADU-S100 in combination with 100 nM ERB, 100 nM VNR, 100 nM CA-4, 100 nM PTX or 500 nM PTX for 6 h normalized to the effect of ADU-S100 alone. (**C**) Analysis of human IFNβ secreted from THP-1 cells treated with 10 μM ADU-S100 in combination with 100 nM ERB, 100 nM VNR, 100 nM CA-4, 100 nM PTX, 500 nM PTX, or a DMSO vehicle control. Significance determined by one-way ANOVA with Dunnett’s posthoc test comparing all conditions to ADU-S100 alone. (**D**) Analysis of human *IFNβ* mRNA in THP-1 cells treated with 10 μM cGAMP in combination with 100 nM ERB, 100 nM VNR, 100 nM CA-4, 100 nM PTX or 500 nM PTX for 6 h normalized to the effect of cGAMP alone. (**E**) Analysis of human *IFNβ* mRNA in HCC1937 cells treated with 33 μM MSA-2 in combination with 100 nM ERB, 100 nM VNR or 500 nM PTX for 6 h. Significance is determined by a one-way ANOVA with Dunnett’s posthoc test comparing all conditions to MSA-2 alone. * *p* < 0.05, ** *p* < 0.01, *** *p* < 0.001.

**Figure 6 cancers-14-05962-f006:**
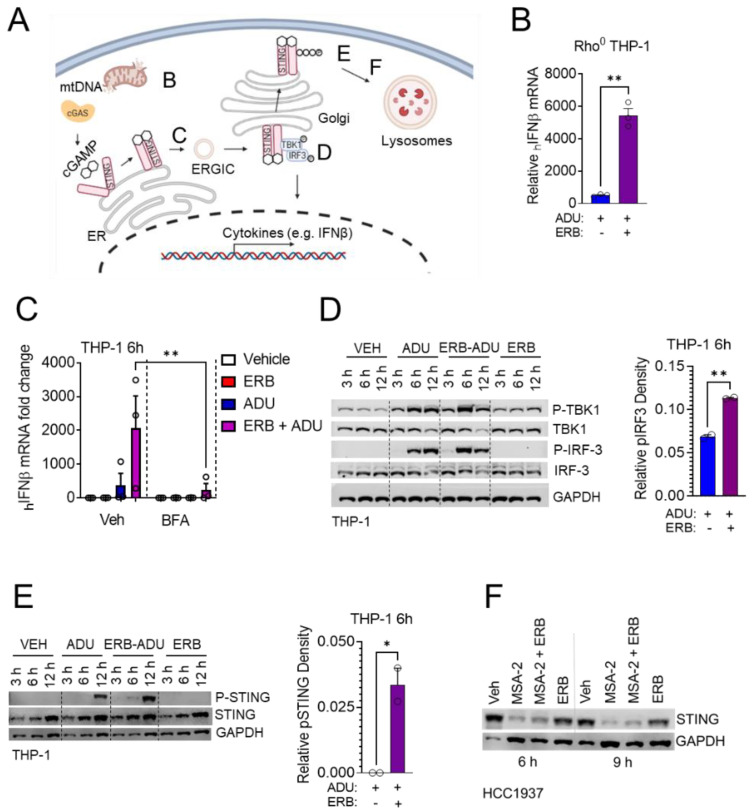
Eribulin increases STING signaling independent of its effects on organelle disruption. (**A**) A schematic diagram depicting the STING signaling pathway. (**B**) Human *IFNβ* mRNA expression in Rho^0^ THP-1 cells treated with 10 μM ADU-S100 (ADU) with or without 100 nM eribulin (ERB) for 6 h. (**C**) Human *IFNβ* mRNA expression in THP-1 cells pretreated with 1 µM brefeldin A (BFA) for 1 h followed by treatment with 10 μM ADU-S100 with or without 100 nM eribulin (ERB) for 6 h. Significance of brefeldin A as determined by a 2-way ANOVA with Sidak’s posthoc test. (**D**) Immunoblot of TBK1 and IRF3 phosphorylation over time in THP-1 cells after ERB and ADU addition and quantification of p-IRF-3 as determined by densitometry of p-IRF-3 over normalized GAPDH at the 6 h time point for all conditions from two independent experiments. (**E**) Immunoblot of STING phosphorylation over time in THP-1 cells after ERB and ADU addition and quantification of p-STING as determined by densitometry of p-STING over total STING at the 6 h timepoint for all conditions from two independent experiments. (**F**) Immunoblot of total STING over time in HCC1937 cells after ERB and MSA-2 addition. Immunoblots are representative of two independent biological replicates. Significance of eribulin shown as determined by 1-way ANOVA with Dunnett’s posthoc test, comparing all conditions to the combination. * *p* < 0.05, ** *p* < 0.01. The uncropped Western blots have been shown in Appendix A.

**Figure 7 cancers-14-05962-f007:**
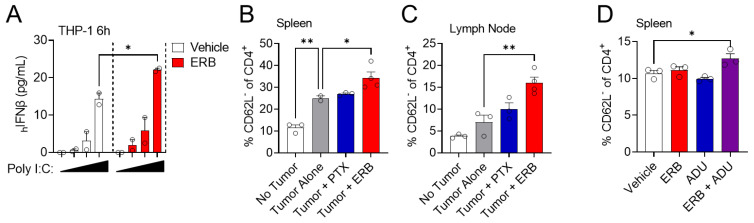
Eribulin enhances the response to multiple immune agonists, in vitro and in vivo. (**A**) Human IFNβ protein secreted from THP-1 cells treated for 6 h with or without 100 nM eribulin in the presence of 0, 1, 2, or 5 µg/mL poly(I:C). Significance of eribulin determined by two-way ANOVA with Sidak’s posthoc test. Frequency of CD62L^−^ CD4^+^ T-cells in (**B**) the spleen or (**C**) the tumor draining lymph node of BALB/c female mice bearing 4T1 tumors compared to no tumor controls or tumored animals receiving 1 mg/kg eribulin Q7Dx2 or 20 mg/kg paclitaxel Q2Dx6 by i.p. injection. Significance determined by one-way ANOVA with Dunnett’s posthoc test comparing all conditions to the tumor alone group. *n* = 2–4 animals/group (**D**) Frequency of CD62L^−^ CD4^+^ T-cells in the spleen of non-tumored FVB/NJ female mice treated with 0.7 mg/kg eribulin Q4Dx5 i.p., 50 µg ADU-S100 Q4Dx3 s.c., or a combination of the two drugs. *n* = 3 animals/group. Percentages of CD62L^−^ CD4^+^ T-cells calculated from the total CD4^+^ T-cell population. Significance was determined by one-way ANOVA with Dunnett’s posthoc test, comparing all conditions to the vehicle control. * *p* < 0.05, ** *p* < 0.01.

**Figure 8 cancers-14-05962-f008:**
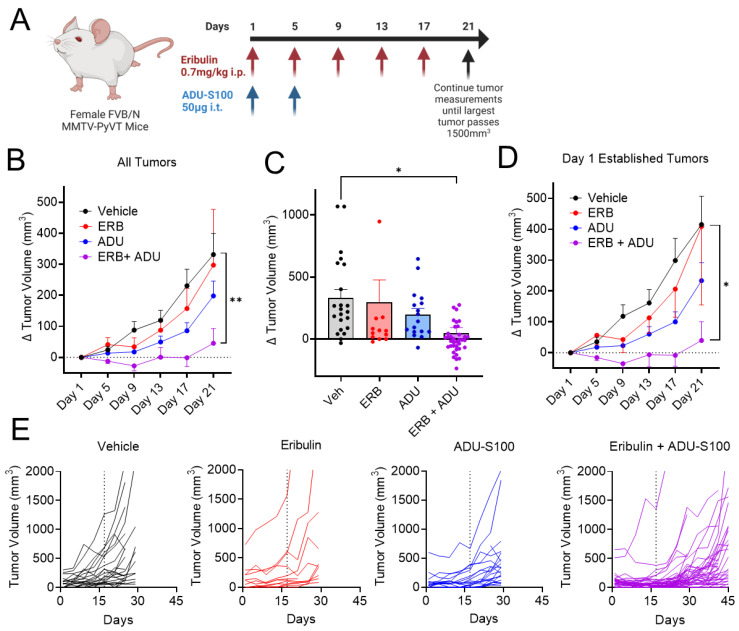
The combination of eribulin with a STING agonist provides anti-tumor efficacy in a transgenic mammary tumor model. (**A**) Timeline of the tumor trial. (**B**) Change in tumor volume of all tumors in each treatment group throughout the dosing period as average ± SEM. (**C**) Individual values for change in tumor volume on day 21. (**D**) Change in tumor volume of only tumors established on day 1 of the tumor trial as average ± SEM. Significance on day 21 shown as determined by mixed-effects analysis (**B**), one-way ANOVA (**C**), or two-way ANOVA (**D**) each with Tukey’s posthoc test. (**E**) Spider plots of individual tumor volumes are shown for each treatment group. ERB + ADU (*n* = 4 mice, 58 tumors), ERB (*n* = 2 mice, 20 tumors), ADU (*n* = 2 mice, 29 tumors), and vehicle (*n* = 2 mice, 28 tumors). Vertical dashed line denotes last day of dosing. * *p* < 0.05, *** p* < 0.01.

## Data Availability

Data is contained within the article and Appendix A.

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
