# Peer review of "The Microtubule Destabilizer Eribulin Synergizes with STING Agonists to Promote Antitumor Efficacy in Triple-Negative Breast Cancer Models"

_cancers, 2022, doi:10.3390/cancers14235962_

Round 1

Reviewer 1 Report (Previous Reviewer 2)

Authors have satisfactorily answered all concerns I had raised.

Reviewer 2 Report (Previous Reviewer 1)

I would like to congratulate the authors for taking my comments very seriously and for their extra work to improve the manuscript. From my perspective, the manuscript is publishable in the present form.

Thank you.

This manuscript is a resubmission of an earlier submission. The following is a list of the peer review reports and author responses from that submission.

Round 1

Reviewer 1 Report

This an interesting work in the field of STING pathway. While the data are encouraging, and the hypothesis may add a benefit for cancer treatment, there are few important things missing in the paper (besides the clear mechanism of synergy between eribulin and STING agonist.

The authors do not rationalize the combination treatment course. Have the authors tested staggered treatments? So far is written in the method, it appears that the treatments were given simultaneously. It will also benefit the paper if the authors will come with an explanation for the observed opposite effects of MSA-2 vs. ADU-S100 or E7766 in HCC1937 vs. HCC1806 cells.

Considering the key event of the main hypothesis is STING activation, the authors must include in their immunoblot the investigation of phospho-STING.

Consequently to STING pathway activation, there is a dramatic STING consumption and degradation in the lysosome (also underlined by the authors, lines 639-640.  All western blot data shows no consumption of STING at 12 hours (Figure 6 D) and even at a very late time point (24 H, figure S3C).

What was the rationale to investigate/show only CD4 CD62 L in the flow cytometry experiments? The methods in section 2.8 shows an extensive panel of antibodies targeting T and B cells. Besides adding the gating strategy, these data should be shown/discussed. How were the CD62L- CD4+ percentages calculated? Are these percentages out of CD4 cell, T cells, or total splenocytes?

Moreover, what was the rationale to not show the anti-tumor effect of the combination in 4T1 model since the flow cytometry was performed in this tumor model.

How many replicates did the authors perform for in vivo data? There is no mention about this in the manuscript.

Figure 5 shows concentrations of IFNb expressed as nanograms/mL. Is this true or is a typo? 500ng/mL is an unusually high value for IFNb.

Sentence from lines 34-36 does not make any sense.

Minor comments in Method section:

Please add the method for mycoplasma detection and cell authenticity.

Please remove ELISA method from section 2.7. It is redundant.

4T1 cells are not mentioned in the method section for cells (section 2.1)

ADU-S100 is soluble in water and is the vendor’s recommended solvent. Any reason for choosing DMSO?

Please revise the “statical” typo in line 330

Reviewer 2 Report

This study by Takahashi-Ruiz et al aims at defining the molecular details of the effect of eribulin, a microtubule destabilizer, on in vitro and in vivo models of triple negative breast cancer (TNBC), in conjunction with agonists of the inflammation mediator STING. Their study tends to demonstrate that eribulin and STING agonists act through a synergistic mode of action, by inducing type I interferon responses. This would help create an inflammatory microenvironment favorable to antitumor activity.

Although this study might be of potential interest in the current context of scarce efficient therapeutic strategies in TNBC, it lacks key control experiments, and thereby fails to establish the validity of the authors' hypothesis.

Major points.
1) No cytotoxicity analyses of any molecules used in this study are presented. This jeopardizes the overall conclusion one could draw from any of the data presented. The data presented in supplemental material do not constitute a cytotoxic analysis.
Additionally, why and how were the actual doses of molecules chosen ? Why 10 uM ADU-S100, while 1 uM E7766 or 33 uM MSA-2 were chosen (Figure 1) ?

2) Since some molecules exert an effect on microtubules (and are known to do so), this effect should be evaluated in a systematic manner on all cell lines used in this study, even though the microtubule-directed activity is not the driver of the anticancer effect. The first evaluation of such activity only comes in Figure 5.

3) For most molecules, the effect of the molecules on IFNb mRNA expression vanishes after 6 to 24h. Why does this occur ? This should be taken into serious account, since authors claim that these molecules are used in clinical settings; so how about any efficacy if the effect vanishes after a few hours of therapy ?

4) Serious concerns about statistical analyses of the data throughout the manuscript. In most figures, only 2 data points are reported, which DOES NOT ALLOW for ANY statistical evaluation. In addition, authors report the evaluation of mean +/- SEM, which is incorrect. Results should be presented as mean +/- SD (standard deviation).

5) Several cell lines are used throughout the study, but often  without consistency. E.g. figure 5 : experiments are conducted with HCC1937 and THP-1 cell lines. Why did the authors choose to compare these 2 ? On top of this, HCC1937 are lymphoblasts (even though isolated from a breast tumor), so they could not be considered as "breast" cancer cells stricto sensu. HCC1806 would have been a better choice here.

6) I have a major problem with the results presented in Figure 3. Panel C : a combination of 100 nM eribulin and 1 uM E7766 is totally unable to activate IFNb production in THP-1 cells. This raises serious doubts about the results presented in Figure 1 : the same combination (100 nM eribulin/1 uM E7766) significantly enhances mRNA IFNb expression in the same cell line, at the same time point. How can authors reconcile such discrepancy ?

Minor points.
1) Figure S3A : what is the link between mitochondrial DNA depletion, and COX1 mRNA expression in Rho0 THP-1 cells ? It is not obvious.

2) How were animals treated, segregated into cages, how were they sacrificed ? This is not mentioned anywhere, and raises concerns about animal welfare.

3) All Y axis where mRNA expression is reported should be expressed as ratio to housekeeping gene (GAPDH ?)

4) More annotations should be placed directly inside the panels, to facilitate the comprehension of the figure (e.g. cell type, doses used...).
